# Contribution of boundary non-stoichiometry to the lower-temperature plasticity in high-pressure sintered boron carbide

Haiyue Xu[1,5], Wei Ji [1,2,5,6] ✉, Jiawei Jiang[2], Junliang Liu [2], Hao Wang[1], Fan Zhang[1,3], Ruohan Yu[4], Bingtian Tu[1], Jinyong Zhang[1], Ji Zou[1], Weimin Wang[1], Jinsong Wu [1,4] & Zhengyi Fu [1,6] ✉

The improvement of non-oxide ceramic plasticity while maintaining the high-temperature strength is a great challenge through the classical strategy, which generally includes decreasing grain size to several nanometers or adding ductile binder phase. Here, we report that the plasticity of fully dense boron carbide ($B_4C$) is greatly enhanced due to the boundary non-stoichiometry induced by high-pressure sintering technology. The effect decreases the plastic deformation temperature of $B_4C$ by 200 °C compared to that of conventionally-sintered specimens. Promoted grain boundary diffusion is found to enhance grain boundary sliding, which dominate the lower-temperature plasticity. In addition, the as-produced specimen maintains extraordinary strength before the occurrence of plasticity. The study provides an efficient strategy by boundary chemical change to facilitate the plasticity of ceramic materials.

Non-oxide ceramics such as boron carbide ($B_4C$) have been widely applied in many advanced structures because of their advantages, which include high hardness, low density, high melting point, excellent chemical stability and wear resistance. However, their application scope is still challenged because of their intrinsic brittleness with limited plasticity, especially within the low to moderate temperature range[1–11].

The lower temperature plasticity of ceramics combined with high strength might improve their in-service reliability through metal-like behavior. Two strategies are generally chosen to improve plasticity at relatively low temperatures[12–15]: (1) decreasing grain size to several nanometers to promote grain boundary sliding; (2) accelerating grain boundary diffusion with low melting point sintering additives. However, it is difficult to sinter dense nanoceramics with grains of less than

10 nm. In addition, secondary phases such as glass and oxide impurities[12,13] in structural ceramics decrease high-temperature strength significantly, thus degrading a key mechanical property. As examples, $Y_2O_3$–$Al_2O_3$–MgO was added to nano-grained $Si_3N_4$ ceramics as a sintering additive, with the as-prepared ceramics exhibiting superplasticity at 1650 °C, with a flow stress of only 4 MPa[12]. Nanocrystalline silicon carbide doped with boron and carbon was found to show a superplastic elongation of >140% at 1800 °C with a yield stress of 80 MPa[13]. In Imamura et al.'s studies about high-temperature deformation of 3Y-TZP, an addition of only 1 wt% amorphous silicate was found to decrease the flow stress by 50% at 1500 °C[14]. Vasylkiv and co-authors investigated the strength promotion of boron carbide and boron carbide-based composites at ultra-high temperature above 2000 °C. The plasticity temperature was elevated as well[7–10]. Zhang

[1]State Key Laboratory of Advanced Technology for Materials Synthesis and Processing, Wuhan University of Technology, Wuhan 430070, China. [2]Department of Materials, University of Oxford, Oxford OX1 3PH, UK. [3]Hubei Longzhong Laboratory, Wuhan University of Technology Xiangyang Demonstration Zone, Xiangyang 441000, China. [4]Nanostructure Research Centre, Wuhan University of Technology, Wuhan 430070, China. [5]These authors contributed equally: Haiyue Xu, Wei Ji. [6]These authors jointly supervised this work: Wei Ji, Zhengyi Fu. ✉e-mail: jiwei@whut.edu.cn; zyfu@whut.edu.cn

et al. found a specific dual-phase α/β-$Si_3N_4$ ceramic with coherent interface, and realized the low temperature plasticity through stress-induced β → α phase transformation at this interface[11].

Recently, it has been proposed that a high density of pre-existing defects can assist the occurrence of low-temperature plasticity in oxide ceramics through a flash sintering method, which could also limit grain growth to obtain fine structures. Via this new strategy, it was found that the enhanced plasticity of $ZrO_2$ and $TiO_2$ at low temperature, ranging from room temperature to 400 °C, was mainly attributed to the high dislocation density, nanoscale stacking faults and point defects formed during the flash sintering process[16,17].

It has been reported that the sintering of ceramics under high pressure with plastic deformation as the dominant sintering mechanism also favors a higher density of defects and further limits grain growth[18-21]. Ji et al. fabricated fully dense $B_4C$ with stacking faults and twins under a pressure of 80 MPa at 1700 °C in 5 min by spark plasma sintering (SPS)[18]. Subsequent studies under a higher pressure of 1.5 GPa further facilitated the fabrication of 3YSZ nanoceramics with a grain size of 60 nm[19]. Very recently, $ZrB_2$ specimens with finer grains were obtained under 15 GPa at 1450 °C, with the high dislocation density and fine grains induced by ultra-high pressure contributing to the enhancement of mechanical and oxidation-resistant properties[20]. Nano-twinned diamond sintered under the condition of 20 GPa and 2000 °C, exhibiting an average twin thickness of 5 nm, achieved unprecedented hardness and stability[21]. Moreover, high-pressure sintering technology could lead to rough grain boundaries[18], alter crystal structure[21] or even chemical equilibrium at grain boundaries, which result in improvement

in the intrinsic properties such as high-temperature yield stress. Therefore, high-pressure sintering has been chosen in the present study as a method of generating better low-temperature plasticity.

In this work, the high-temperature mechanical properties of $B_4C$ ceramics fully densified by high-pressure low-temperature (HPLT) sintering technology and by conventional low-pressure high-temperature (LPHT) method are explored, respectively. $B_4C$ is an interesting material for various applications, including light-weight armor, cutting tools and nuclear reactor components[22-25]. Additionally, as a non-stoichiometric ceramic, $B_4C$ has a highly tunable B:C ratio, which suggests a promising opportunity to "customize" its intrinsic properties by breaking the chemical equilibrium of 4:1[26-28]. The microstructure evolutions, especially the grain boundary atomic distribution, are analyzed. Compared to reference specimens sintered under the LPHT condition, the HPLT-$B_4C$ ceramics exhibit plasticity at much lower temperatures, which is considered to relate to the boundary non-stoichiometry.

## Results

### Flexural behavior at elevated temperatures

Figure 1 shows the flexural deformation behavior and the trend from brittle behavior to ductility as the testing temperature increased. The temperature dependence of flexural strength of both HPLT-$B_4C$ and LPHT-$B_4C$ firstly increased with temperature prior to decreasing again as plasticity began to occur. The flexural strength of HPLT-$B_4C$ increased from 608 MPa to 822 MPa within the range of 110–1400 °C and was still higher than 600 MPa before significant plasticity occurred at 1600 °C. After plasticity occurred, the strength reduced to 385 MPa

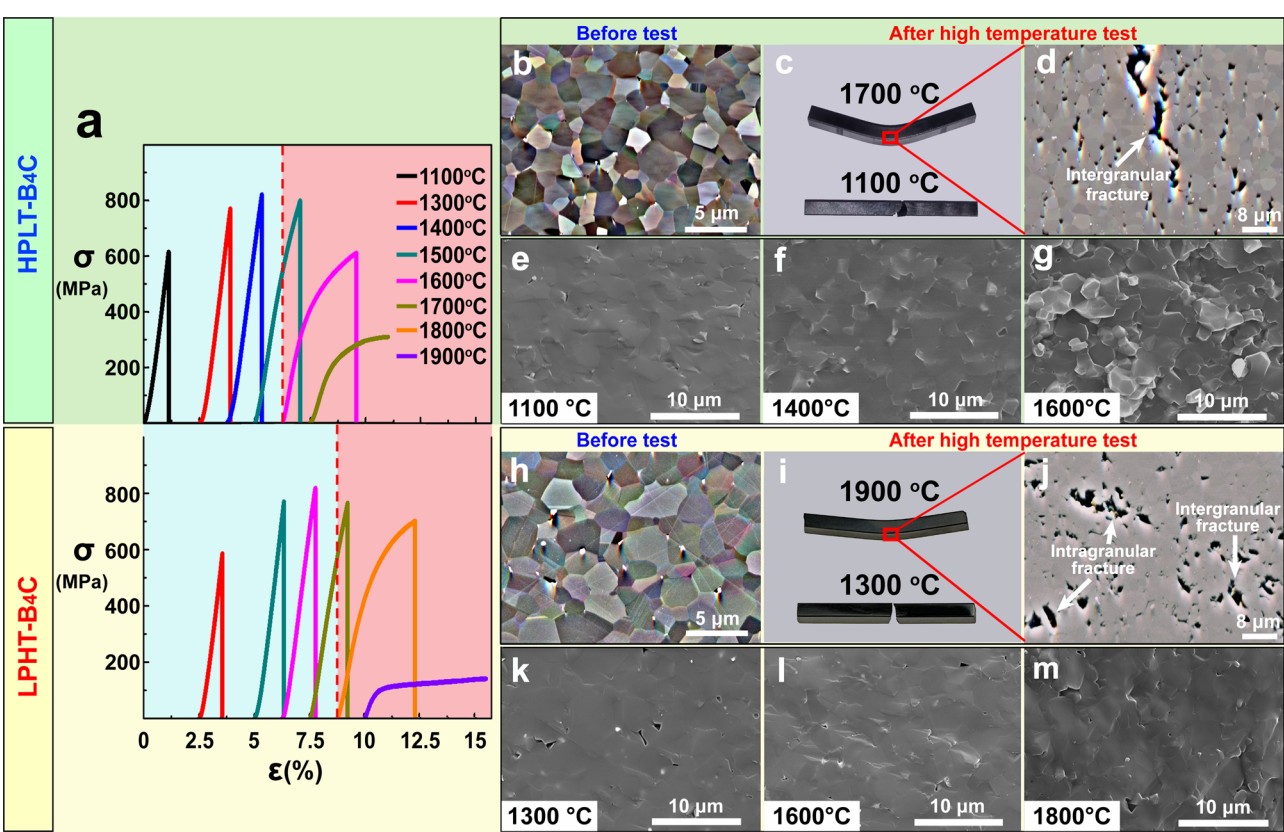

**Fig. 1 | High-temperature flexural test with $B_4C$ ceramics at a constant strain rate of 5.7 × 10⁻⁴ s⁻¹. a** The upper and lower parts respectively depict the high-temperature flexural stress-strain behavior of HPLT-$B_4C$ and LPHT-$B_4C$. The curves indicate the tests of HPLT-$B_4C$ at 1100–1700 °C and the tests of LPHT-$B_4C$ at 1300–1900 °C. **b, h** Forescattered electron images of as-produced HPLT-$B_4C$ and LPHT-$B_4C$. **c, i** Photos of HPLT-$B_4C$ and LPHT-$B_4C$ after high-temperature flexure at 1100 °C, 1700 °C and 1300 °C, 1900 °C, respectively. **d, j** Morphologies of regions under maximum strain during the test shown in **c** 1700 °C specimen and **i** 1900 °C specimen. SEM images of tensile fracture surface of $B_4C$ after high-temperature flexural test at different temperatures: **e** 1100 °C, **f** 1400 °C, **g** 1600 °C for HPLT-$B_4C$ and **k** 1300 °C, **l** 1600 °C, **m** 1800 °C for LPHT-$B_4C$. Source data are provided as a Source Data file.

**Table 1 | Summary of sintering, test conditions and flexural strength of B$_4$C ceramics**

| Samples | Sintering temperature (°C) | Sintering pressure (MPa) | Test temperature (°C) | Crosshead speed (mm/min) | Flexure strength (MPa) |
|---|---|---|---|---|---|
| HPLT-B$_4$C-1100-0.5 | 1800 | 80 | 1100 | 0.5 | 617 |
| HPLT-B$_4$C-1300-0.5 | 1800 | 80 | 1300 | 0.5 | 772 |
| HPLT-B$_4$C-1400-0.5 | 1800 | 80 | 1400 | 0.5 | 823 |
| HPLT-B$_4$C-1500-0.5 | 1800 | 80 | 1500 | 0.5 | 801 |
| HPLT-B$_4$C-1600-0.5 | 1800 | 80 | 1600 | 0.5 | 612 |
| HPLT-B$_4$C-1700-1.0 | 1800 | 80 | 1700 | 1.0 | 454 |
| HPLT-B$_4$C-1700-0.5 | 1800 | 80 | 1700 | 0.5 | 320 |
| HPLT-B$_4$C-1700-0.25 | 1800 | 80 | 1700 | 0.25 | 232 |
| HPLT-B$_4$C-1800-0.5 | 1800 | 80 | 1800 | 0.5 | 130 |
| HPLT-B$_4$C-1900-0.5 | 1800 | 80 | 1900 | 0.5 | 39 |
| HPLT-B$_4$C-2000-0.5 | 1800 | 80 | 2000 | 0.5 | 17 |
| LPHT-B$_4$C-1300-0.5 | 2100 | 20 | 1300 | 0.5 | 588 |
| LPHT-B$_4$C-1500-0.5 | 2100 | 20 | 1500 | 0.5 | 772 |
| LPHT-B$_4$C-1600-0.5 | 2100 | 20 | 1600 | 0.5 | 821 |
| LPHT-B$_4$C-1700-0.5 | 2100 | 20 | 1700 | 0.5 | 767 |
| LPHT-B$_4$C-1800-0.5 | 2100 | 20 | 1800 | 0.5 | 703 |
| LPHT-B$_4$C-1900-0.5 | 2100 | 20 | 1800 | 0.5 | 143 |
| LPHT-B$_4$C-1950-1.0 | 2100 | 20 | 1950 | 1.0 | 101 |
| LPHT-B$_4$C-1950-0.5 | 2100 | 20 | 1950 | 0.5 | 71 |
| LPHT-B$_4$C-1950-0.25 | 2100 | 20 | 1950 | 0.25 | 55 |
| LPHT-B$_4$C-2000-0.5 | 2100 | 20 | 2000 | 0.5 | 36 |

at 1700 °C[29,30]. The flexural strength of LPHT-B$_4$C increased from 580 MPa to 821 MPa when the temperature increased from 1300 to 1600 °C[10]. Significant plasticity first occurred at 1800 °C and the flexural strength decreased to 190 MPa at 1900 °C. At the initial stage, the increase in flexural strength with temperature within a moderate temperature range might be ascribed to the high temperature-activated relaxation of thermal residual stress[31,32]. However, after the brittle-ductile transition temperature was reached, the plastic yield stress decreased with elevated temperatures[7–10,29,30].

The photographs and microstructures of HPLT-B$_4$C and LPHT-B$_4$C after flexural deformation at 1700 °C and 1900 °C are shown in Fig. 1c, d and i, j. At high temperature, both HPLT-B$_4$C and LPHT-B$_4$C specimens exhibited a similar plastic morphology to that observed with metals (Fig. 1c, i). The brittle-ductile transition of HPLT-B$_4$C and LPHT-B$_4$C occurred at 1600 °C and 1800 °C respectively. Substantial plastic deformation of HPLT-B$_4$C and LPHT-B$_4$C occurred at 1700 °C and 1900 °C respectively, revealing that the plastic deformation temperature of HPLT-B$_4$C was 200 °C lower than that of LPHT-B$_4$C. This difference in temperature of ceramic plasticity may be related to relative density, grain size, internal stress, micro-morphology and grain boundary structure[33–35]. Both materials possessed relative densities above 99%. The average grain sizes of HPLT-B$_4$C and LPHT-B$_4$C were similar, 2.5 μm and 3.1 μm respectively (Fig. 1b, h). Therefore, the plasticity at lower temperatures was not mainly attributed to the relative density or grain size. Although previous studies have ascribed the variation of high-temperature plasticity to activated relief of thermal residual stresses[31], the testing results of annealed samples in the present study (Supplementary Fig. 1 and Supplementary Table 1) indicated that the plastic deformation temperature was independent of residual stresses. Therefore, the decrease in plastic deformation temperature might be related to special micro-morphology and grain boundary structure formed by the high-pressure low-temperature sintering technology.

As for HPLT-B$_4$C-1700-0.5, named and listed in Table 1, with the increase in the strain during the high-temperature plastic deformation, the rough failure area induced by microcrack propagation was observed in the surface center where under the maximum tensile strain during the test (Fig. 1d, g). The location was determined from the

finite element analysis as shown in Supplementary Fig. 2. Intergranular fracture was detected clearly near the crack tip (Fig. 1d) and a high density of pores was evident on the surface. The high-temperature fracture of B$_4$C was induced by nucleation, propagation, blunting and coalescence of cracks. These pores could favor the micro-void coalescence ahead of the crack tips. Plasticity at this temperature may relieve stress at the crack tips, facilitating the growth and coalescence of cracks[33,36]. The morphology revealed that intergranular fracture was the dominant failure mode of HPLT-B$_4$C in the present work, while LPHT-B$_4$C-1900-0.5 behaved differently in terms of fracture mode and morphology. The fracture occurred both at the grain boundary and inside grains (Fig. 1j, m), indicating that the failure was caused by both transgranular and intergranular fractures. It is important for these different fracture modes of the two B$_4$C to be further investigated, as the mechanism may directly lead to the lower-temperature plasticity of the HPLT-B$_4$C.

Despite the different fracture modes of HPLT-B$_4$C and LPHT-B$_4$C as stated above, neither of them is similar to metallic materials which generally exhibit elongated grains after deformation. The electron back scattered diffraction (EBSD) results in Supplementary Figs. 3 and 4 show that the grains of the B$_4$C specimens, either with or without deformation, were equiaxed.

## Boundary non-stoichiometry of high-pressure sintered B$_4$C

It has been reported that the intrinsic properties of B$_4$C at grain boundaries or within the grains are altered by breaking the stoichiometric ratio of 4:1[27]. Therefore, the stoichiometry and crystal structure of grain lattice and grain boundary in as-sintered B$_4$C (without high-temperature deformation) are investigated here.

Electron energy loss spectroscopy (EELS) spectra were used as a preliminary method to study the B$_4$C structure along the grain boundary and within the grains. The EELS spectra in Fig. 2a, b showed identical structures on grain boundaries and within the grains in LPHT-B$_4$C. However, this was not the case for HPLT-B$_4$C. As shown in Fig. 2g, h, EELS spectra show a transition from multiple, discrete peaks in the range of 193–205 eV in the lattice of HPLT-B$_4$C, to a single smooth curve in the grain boundary. This trend in EELS spectra was consistent

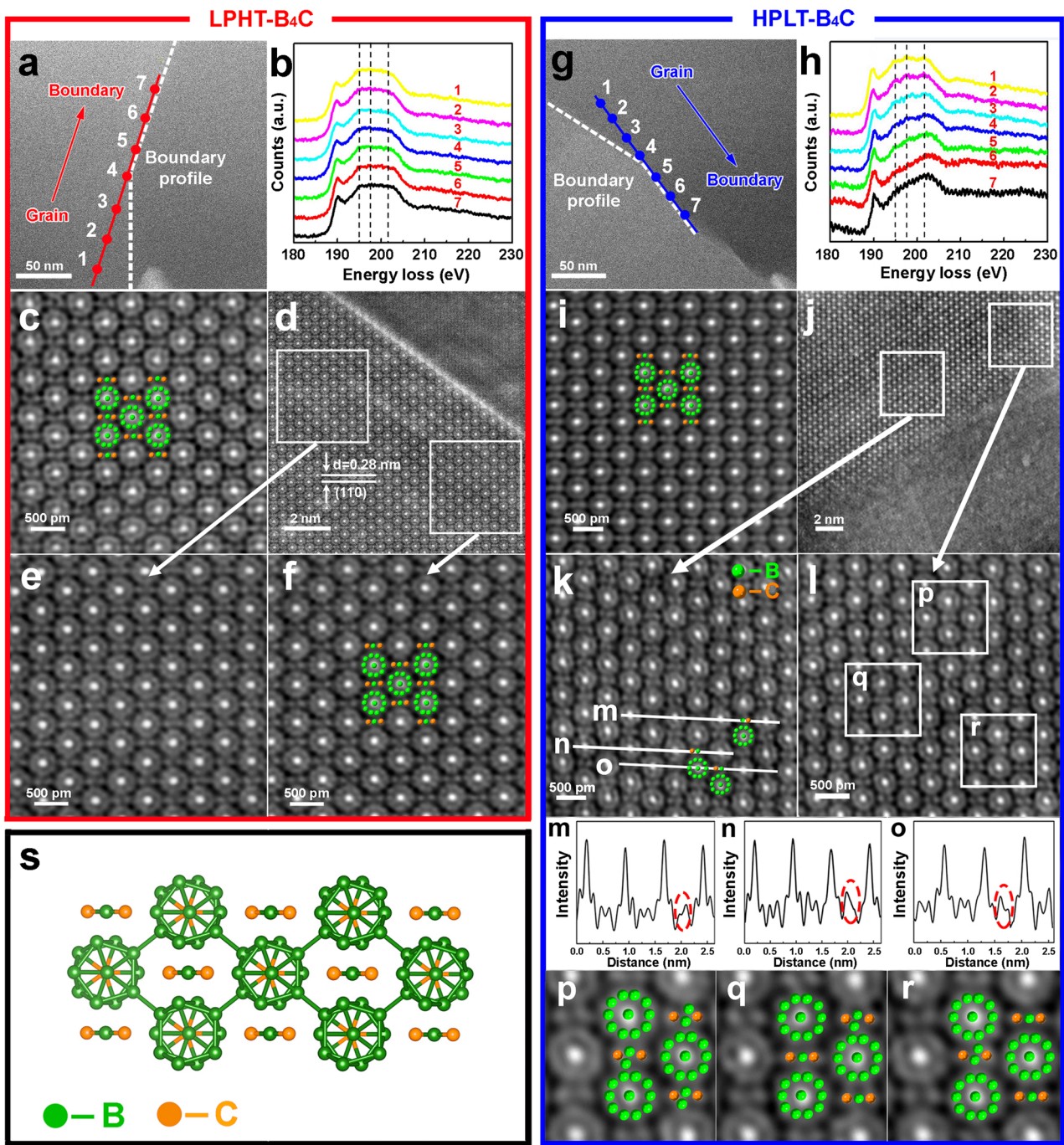

**Fig. 2 | Detailed microstructure analysis of the grain lattice and grain boundary in the as-sintered B$_4$C ceramics.** EELS analysis (**a**, **b**, **g**, **h**) and spherical aberration corrected TEM analysis (**c**–**f**, **i**–**r**) of as-sintered **a**–**f** LPHT-B$_4$C and **g**–**r** HPLT-B$_4$C at various locations from inner grain to grain boundary. **a**, **g** Boundary morphology. **b**, **h** EELS results of corresponding points in (**a**, **g**). The peak change in (**h**) indicates the boundary non-stoichiometry. **c** Atomic arrangements of inner grain. **d** Grain boundary morphology. **e**, **f** Atomic arrangements near the grain boundary. No

obvious change can be observed in LPHT-B$_4$C. **i** Local atomic arrangement of inner grain. **j** Grain boundary morphology. **k**, **l** Atomic arrangements along the grain boundary. **m**–**o** Carbon deficiency along the boundary corresponding to (**k**). **p**–**r** Boron enrichment near the boundary corresponding to (l), reflecting the bent chains CBC with different orientations of singular chains, linear chain combined with rhombus CB$_2$C and rhombi CB$_2$C, respectively. **s** Atomic model of perfect B$_4$C. Source data are provided as a Source Data file.

with that reported by Xie et al. with the increase in B/C ratio beyond 4:1, confirming the chemical element non-stoichiometry at the grain boundary of HPLT-B$_4$C[27].

To further confirm the unique crystal structure, a spherical aberration corrected transmission electron microscope (ACTEM) study was performed at the grain boundary and lattice area of both B$_4$C ceramics (Fig. 2c–f, i–r). Atomic models of the ideal B$_4$C crystal structure in the [211] crystallographic direction yielded the best view[37].

The icosahedra and the chains were projected onto each other. The conventional high-temperature sintering method led to a straight equilibrium grain boundary (Fig. 2d). LPHT-B$_4$C ceramics showed consistent chemical bonding and atomic occupation inside the grain and at the grain boundary (Fig. 2c and e, f), exhibiting perfect structures of icosahedral clusters and CBC chains. The grain boundary and inner grains of B$_4$C ceramics had the same intrinsic structures and properties, thus leading to the coexistence of transgranular and

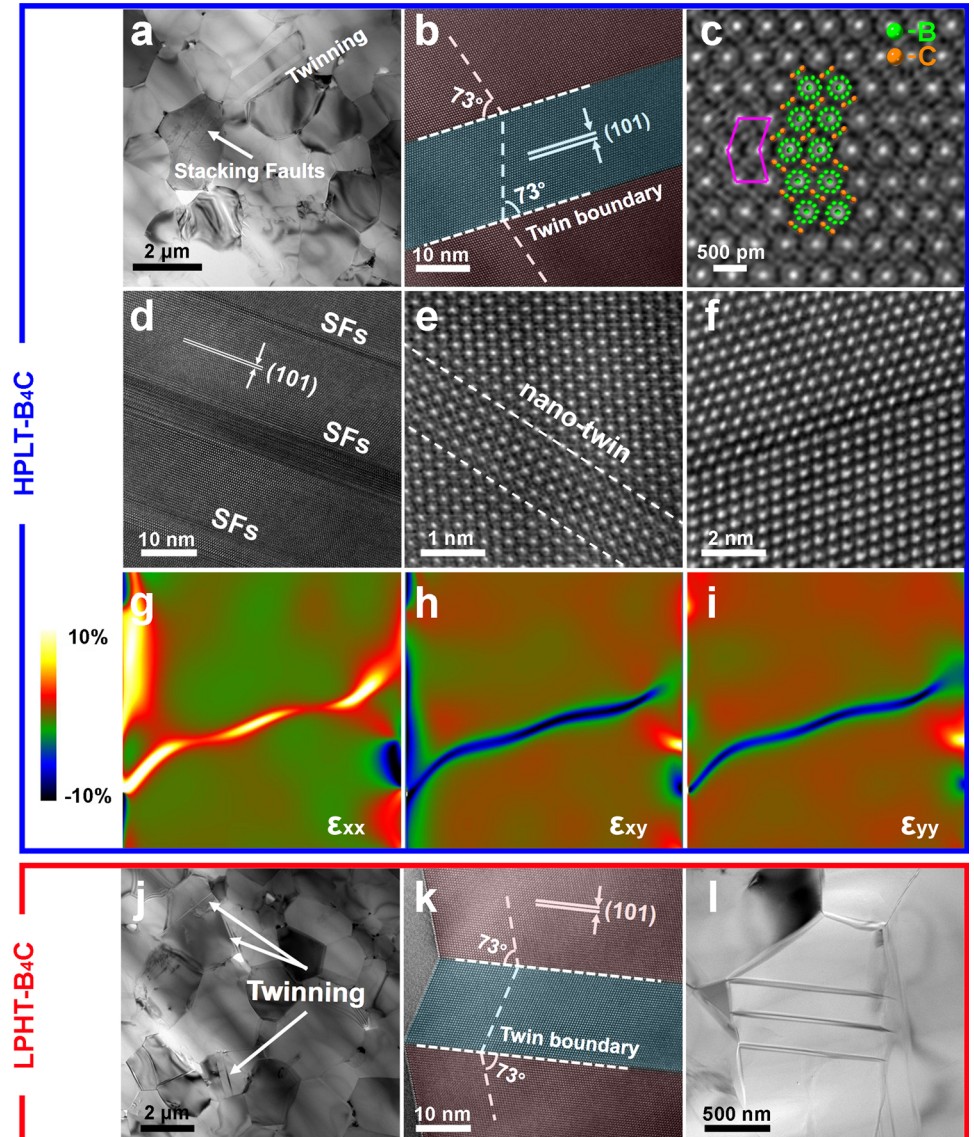

**Fig. 3 | TEM analysis of the high-temperature flexural deformed B₄C ceramics.** The results of the samples initially sintered respectively under the conditions of **a–i** high pressure and low temperature and **j–l** low pressure and high temperature. **a** Bright-field TEM micrograph. **b–e** HRTEM images of large-sized twins, twin boundary, nanoscaled stacking faults and twins. **f–i** Typical twin boundary and the corresponding strain along the horizontal ($\varepsilon_{xx}$), shear ($\varepsilon_{xy}$), and vertical ($\varepsilon_{yy}$) directions. **j** Bright-field TEM micrograph. **k** HRTEM images of large-sized twinning. **l** Bright-field TEM micrograph of large-sized twinning without nano-twins and stacking faults.

intragranular fractures in the flexure test at 1900 °C, as indicated by the fracture mode shown in Fig. 1.

The atomic arrangement of inner lattice of HPLT-B₄C (Fig. 2i) also exhibited the same perfect B₄C structure as that of LPHT-B₄C. However, high resolution TEM images (Fig. 2j) indicated that the non-stoichiometry rough grain boundary formed during high-pressure sintering. The atomic images (Fig. 2m–o) clearly confirmed that the high-pressure and low-temperature sintering technology could induce the vacancy of carbon in the CBC chains in the grain boundary yielding region. Meanwhile, the atomic configuration of boron carbide with higher B concentrations along grain boundary was also observed (Fig. 2p–r). Strong atomic arrangement distortion was observed along the grain boundaries, like the Rasim model[26]. Both B concentration and C atom vacancy induced a larger B/C atomic ratio of the crystal structure near grain boundaries than inner grains.

The result illustrated direct evidence that the high-pressure and low-temperature sintering technology could alter the chemical composition at grain boundaries and may affect the grain boundary performance of B₄C ceramics.

**Plastic deformation characterization in the tensile surface**
Post-deformation TEM analysis with B₄C ceramics after high-temperature flexural tests was performed in the present study. A TEM bright field image of HPLT-B₄C (Fig. 3a) shows the formation of high-density defects including large-sized twins, stacking faults and nano-twins. In Fig. 3b, the angle between matrix planes and twins is 73°. Figure 3c shows a scanning transmission electron microscopy (STEM) image collected by a high-angle annular dark-field (HAADF) detector in ACTEM. Based on the arrangement of the icosahedral clusters (which are the unique atomic chains in the rhombohedral structure), it is revealed that the two crystals separated by the interface have a perfect mirroring relationship while the inclination angle is 73°. As shown in Fig. 3d, e, in addition to large-sized twins, nano-twins were also observed in HPLT-B₄C-1900-0.5 grains. According to the TEM images in Supplementary Fig. 5, nano-twins with around 1-2 nm width had

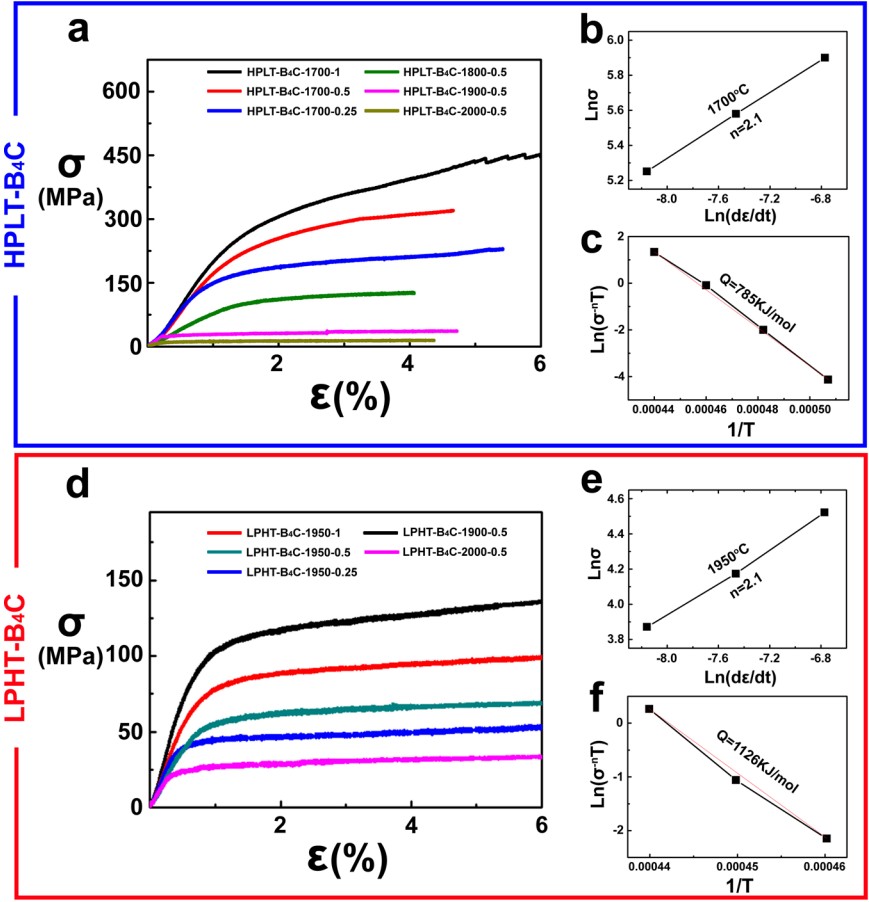

**Fig. 4 | High-temperature plastic deformation behavior and mechanisms of the deformed B$_4$C ceramics. a–c** HPLT-B$_4$C and **d–f** LPHT-B$_4$C. **a, d** Stress−strain curves, **b, e** strain rate dependence of yield stress and **c, f** calculation lines of activation energy for high-temperature flexural tests at elevated temperatures. Source data are provided as a Source Data file.

preexisted after high-pressure sintering prior to high-temperature flexural tests.

By contrast, in B$_4$C ceramics sintered by conventional high-temperature method, with 150 grains carefully investigated, it was found that only large-sized twins without nanoscale defects could be observed (Fig. 3j–l), which might also indicate that the nano-twins in deformed HPLT-B$_4$C initially formed during high-pressure sintering rather than deformation. In addition, as indicated in EBSD results shown in Supplementary Fig. 4, the density of large-sized twins basically remained constant after high-temperature plastic deformation.

Neither the large-sized twins nor nano-twins directly led to the change in fracture modes and dominated the lower-temperature plasticity, but might still make a positive contribution to the high-temperature plastic deformation.

Here geometric phase analysis was applied as an indirect way to confirm the effect of twinning on the force field. The results based on the (101) lattice reflections of a 73°<211> twin interface inside the HPLT-B$_4$C-1900-0.5 after flexural deformation are shown in Figs. 3f and 4i, where ε$_{xx}$, ε$_{xy}$, and ε$_{yy}$ are horizontal axis strain, shear strain and vertical axis strain, respectively[38,39]. The local strain fields showed tensile strain (orange color) along the horizontal axis and compressive strain (blue color) with the vertical axis component. There is also significant shear strain. The normal strain tensors were calculated as a symmetric matrix so as to relieve lattice mismatch at the twin plane. The fracture and large strain of B$_4$C ceramics are associated with the twin interface[27,38,40]. The compressive strain exhibited that, due to the improvement in shear resistance, the 73°<211> twins in B$_4$C presented a positive influence on strength, thus maintaining the mechanical

properties at relatively high temperatures of ~1600 °C. The Raman results shown in Supplementary Fig. 6 and Supplementary Table 2 confirmed that the structure of B$_4$C ceramics did not become amorphous after high-pressure and high-temperature deformation, demonstrating the stability of the B$_4$C ceramics at high temperatures, which is different from the crystal change at room temperature failure[41,42].

## Discussion

To further investigate the plastic deformation mechanism, typical temperatures of 1700 °C and 1950 °C for HPLT-B$_4$C and LPHT-B$_4$C respectively were selected in the study. The dependence of the diffusion-controlled deformation kinetics on temperature, stress and strain rate can be expressed as follows[43]:

$$\dot{\varepsilon} = \frac{ADGb}{T}\left(\frac{\mathbf{b}}{d}\right)^p \left(\frac{\sigma}{G}\right)^n e^{-\frac{Q}{RT}} \qquad (1)$$

where $\dot{\varepsilon}$ is the strain rate; $A$ is a constant; $D$ is the diffusion coefficient; $\mathbf{b}$ is the Burgers vector; $T$ is the temperature; $d$ is the grain size; $p$ is the grain size exponent; $\sigma$ is the stress; $G$ is the shear modulus; $Q$ is the activation energy; R is the gas constant; $n$ is the stress exponent and is used to identify the dominant mechanism.

Based on the constitutive equations, assuming that the temperature is constant, the relation between the stress $\sigma$ and strain rate $\dot{\varepsilon}$ is

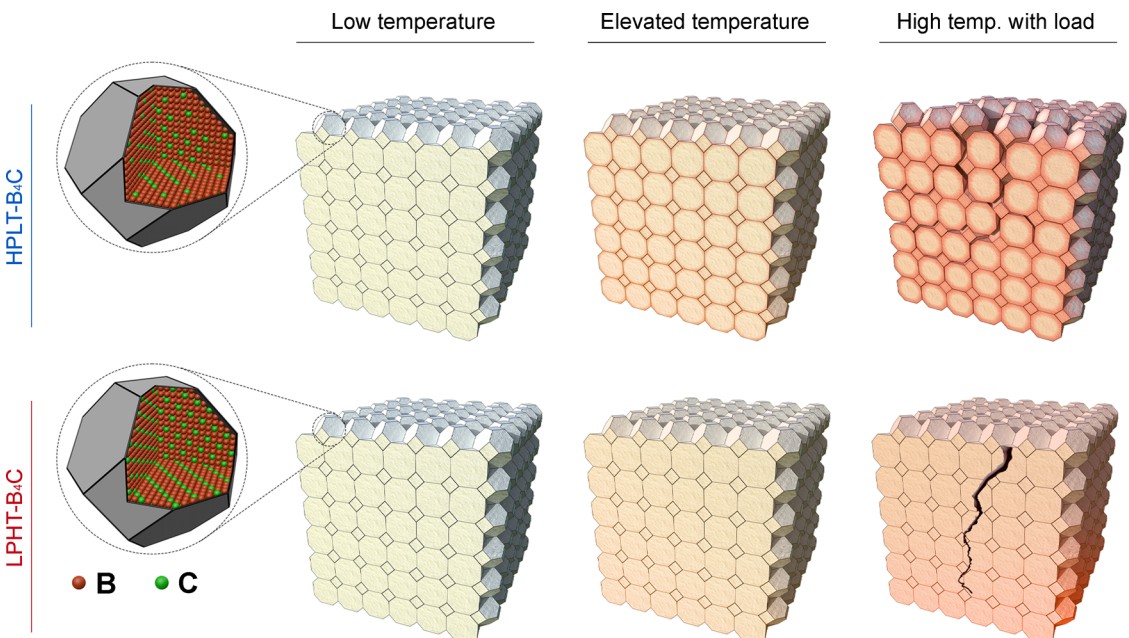

**Fig. 5 | Diagram of boundary non-stoichiometry's dominant contribution on the different fracture modes at rising temperature upon loading.** Lower B/C atomic ratio in the HPLT-B₄C results in more intergranular fracture and lower-temperature plasticity.

simplified as[44]:

$$\sigma = B\dot{\varepsilon}^{\frac{1}{n}} \qquad (2)$$

where B is a constant.

The activation energy Q was determined from the slope of the line shown in Fig. 4(c) and (f), which plot $\ln(\sigma^n T)$ at the same strain rate as a function of $1/T$ in the high-temperature deformation process.

The stress exponents $n$ were calculated with the stress-strain curve obtained under various strain rates and a constant temperature. The activation energy Q was determined from the stress–strain curves obtained under different temperatures and a constant strain rate. The results showed that the plastic yield stress increased with the strain rate at a constant temperature.

HPLT-B₄C and LPHT-B₄C exhibited identical stress exponents ($n = 2.1$), suggesting that the plastic deformation of HPLT-B₄C and LPHT-B₄C was dominated by grain boundary sliding[40]. This is consistent with the observed intergranular fracture more in HPLT-B₄C when significant plasticity occurred. The values of stress exponent in the present work ($n = 2.1$) were smaller than that from the available literature ($n = 3$), which was associated with deformation by dislocation motion[29]. The main reason was that the specimens in the current study possessed much smaller grain size, which favored grain boundary sliding with $n = 2$ rather than power law creep with $n = 3$. In addition, the heavy twinning introduced during processing of the HPLT-B₄C would suppress dislocation motion[45]. Moreover, the present study mainly focused on the plastic deformation rather than creep conditions, with the strain rate of high-temperature deformation adopted here (within the range of $2 \times 10^{-4} \sim 8 \times 10^{-4}\,\text{s}^{-1}$) being much higher than that in previous studies ($10^{-6}\,\text{s}^{-1}$).

The values of Q for HPLT-B₄C and LPHT-B₄C were respectively 785 kJ/mol and 1126 kJ/mol, suggesting an acceleration of the dominant diffusion mechanism of HPLT-B₄C relative to LPHT-B₄C. Therefore, it indicated that the high-pressure and low-temperature sintering technology promoted grain boundary diffusion-controlled plasticity at a relative low temperature. The similar strengths and fracture modes of the two materials in the brittle fracture regime indicates that the grain boundaries were not significantly weakened by the HPLT treatment.

The activation energy values calculated from flexural deformation in the present study were higher than those in previous studies, which ranged from 385 kJ/mol to 632 kJ/mol[29,30]. Effects of different sintering technologies, grain sizes measurement parameters and plastic deformation mechanism could be responsible for this increase in activation energy. As noted above, the dominant deformation mechanism, the strain-rate regime and the microstructures are significantly different in this work to the previous studies.

The above investigations revealed that the dominant mechanism of high-temperature plastic deformation was grain boundary sliding, rather than a direct effect of defects in nanoscale or larger scale.

In a pressure-assisted sintered B₄C sample, both asymmetric and symmetric twins were detected (Supplementary Fig. 5), consistent with the atomic ratio of B to C in the grain lattice (4:1)[46], while the observed non-stoichiometry resulted in a larger B:C ratio at grain boundaries. According to the previous study[27], when the atomic ratio of B to C increased above 4:1, the intrinsic hardness slightly decreased. Therefore, the hardness of HPLT-B₄C along grain boundaries was correspondingly lower than that inside the grains. The yield stress of B₄C ceramics at high temperatures was positively correlated with the hardness because there was no obvious grain deformation or texture during plastic deformation[32,47]. The yield stress of dense ceramics is estimated to be a third of the hardness. Therefore, the grain boundary diffusion was promoted. As shown in Fig. 5, the plastic yielding of HPLT-B₄C started at grain boundaries with slightly lower yield stress, leading to intergranular fractures. In contrast, for LPHT-B₄C, the crack needed higher temperature to conquer the relative higher yield stress, which was identical between grain and boundary, and resulted in the mixture of transgranular and intergranular fractures. Ultimately, it was revealed that the unique crystals with boundary non-stoichiometry formed by high-pressure and low-temperature sintering technology greatly contributed to the improvement in plasticity at lower temperatures.

In summary, flexure experiments at high temperatures showed that B₄C sintered under a high pressure had the better low-temperature plasticity. The plastic deformation temperature of the B₄C ceramics sintered with high-pressure sintering technology was 200 °C lower than that of B₄C ceramics fabricated by conventional high-temperature sintering, with the plastic deformation mainly

dominated by grain boundary sliding. Boundary non-stoichiometry in HPLT-B$_4$C decreased the activation energy and thus favored lower-temperature plastic deformation. By regulating the state of the grain boundaries, the high-pressure sintering technology could stimulate the lower-temperature plasticity of ceramics while maintaining high strength before the plasticity occurred. The technology proposed in the present study could enable non-oxide ceramics to meet the engineering application requirements in a broader range of temperatures.

## Methods

### Sintering of B$_4$C

As-received B$_4$C powder (97% purity, Mudanjiang Diamond Boron Carbide Co., Ltd., China) with a median particle size of 2.5 μm was used as the starting material. The impurities include 0.7% N, 1.8% O, 0.1% Fe, 0.15% Si, 0.05% Al and 0.2% other elements (Ca, Cr, Mg, Mn, Ni, Ti, W). B$_4$C ceramics were sintered by SPS under two different conditions: (i) 80 MPa and 1800 °C for HPLT-B$_4$C, and (ii) 20 MPa and 2100 °C for LPHT-B$_4$C. For both conditions, 22 g raw powder was first poured into a cylindrical graphite die with an inner diameter of 50 mm, prior to the following sintering process in a SPS apparatus (HPD 60, FCT, Germany). and the temperature increased gradually from room temperature to the desired temperature with a heating rate of 100 °C/min, and then maintained for 10 min. The sintering pressure was applied during the entire heating process. After the completion of the above heating process, the pressure was immediately released to 10 MPa at the beginning of the cooling process. In order to minimize the effects of residual stresses, the temperature first decreased to 1600 °C at a constant cooling rate of 10 °C/min, followed by natural cooling to room temperature.

### High-temperature flexural tests

Test bars (2.2 mm × 3.8 mm × 25 mm) for three-point flexural measurements at elevated temperatures were cut from the sintered pellets using diamond saws. The high-temperature three-point flexural strengths were measured by a ceramic test system with New Equipment Machinery Systems (NEMS) in a high temperature vacuum furnace (MTS, AGS-X, SHIMADZU, Japan) with a span of 16 mm according to literatures and the Chinese Standard GBT 14390-2008 (corresponding to ASTM C1211)[48–50]. The stress and strain were calculated from the loading and displacement data using Eqs. (3) and (4)[48]. The strain rate was estimated from the crosshead speed according to Eq. (5). The test conditions of HPLT-B$_4$C and LPHT-B$_4$C are listed in Table 1.

$$\sigma = \frac{5FL}{4ac^2} \quad (3)$$

$$\varepsilon = \frac{8cl}{L^2} \quad (4)$$

$$\dot{\varepsilon} = \frac{8cv}{L^2} \quad (5)$$

where $F$ is the applied load; $L$ is the outer span (16 mm); $a$ and $c$ are the width and thickness of the test bar; $l$ is the crosshead displacement; $v$ is the crosshead speed. A minimum of three specimens were measured at each test temperature.

### Microstructural characterization

BSE images (Fig. 1b, d, h, i) were recorded with a field emission scanning electron microscope (FESEM, Merlin, Zeiss, Germany) and taken combined with the signals from a multi-detector system (Bruker ARGUS ™), which was positioned below the phosphor screen to collect forward scattered electrons. Detailed microstructures (Fig. 3a, b, d, f–l) were examined using high resolution transmission electron microscopy (HRTEM, Talos-F200S, FEI, USA), with the atomic-scale characterization (Figs. 2c–f, i–l, p–r, and 3c, e,) performed in a double spherical aberration corrected transmission electron microscope (ACTEM, Titan Cubed Themis G2 300, FEI, USA). The B and C elemental concentrations from grain boundary to within the grain (Figs. 2a, b, g, h) were investigated from the chemical bonding information using electron energy loss spectroscopy (EELS, GIF QuantumER) in the TEM. TEM specimens were prepared from as-processed B$_4$C by mechanical thinning and subsequent ion milling (EM TXP, Leica, Germany).

The strain distribution (Supplementary Fig. 2) during the bending process under quasi-static loading was determined by finite element analysis. The numerical simulation was performed with bilinear elastoplastic mechanical model based on the intrinsic natures of the indenters and test bar. The test bar was regarded as a plastic body at high temperatures. TEM specimens were used to explore the morphology evolution of B4C before and after high-temperature deformation (Supplementary Figs. 3 and 4, respectively) using scanning electron microscope-transmission Kikuchi diffraction (SEM-TKD, JSM-7500F, JEOL, Japan) with an electron back scattered diffraction (EBSD) detector (Symmetry EBSD, Oxford Instrument, UK). The atomic-scale characterization (Supplementary Fig. 5) of the asymmetric nano-twins in HPLT-B4C was performed via a double spherical aberration corrected transmission electron microscope (ACTEM, Titan Cubed Themis G2 300, FEI, USA). The fully plastic deformation was carried out in the whole process. The Raman spectra (Supplementary Fig. 6) of the high-temperature plastic deformed B4C ceramics were recorded with a Raman microscope (Renishaw, UK) with the excitation laser line at 532 nm. The peak at 1092 cm$^{-1}$ corresponding to the stress-induced shifting was identified by fitting with a mixture of Gaussian and Lorentzian functions[51].

## Data availability

All other relevant data are available from the corresponding author upon reasonable request. Source data are provided with this paper.

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

## Acknowledgements

The authors thank Prof. Richard Todd from University of Oxford for the academic guidance in deformation mechanism and language polish. The authors acknowledge the use of characterization facilities within the David Cockayne Centre for Electron Microscopy, Department of Materials, University of Oxford. The authors also thank Prof. Danyu Jiang and Prof. Xingang Wang from Shanghai Institute of Ceramics, Chinese Academy of Sciences for the help in high-temperature flexural strength testing. W.J., W.M.W. and J.Z. acknowledge the support of National Natural Science Foundation of China (Grant Nos. 92163208, 52322207, 52022072, 51902233). W.M.W. also thanks the support of National Key Research and Development Plan of China (2021YFB3701400). W.J. also acknowledges the financial support of Independent Innovation Projects of the Hubei Longzhong Laboratory (2022ZZ-11). The ACTEM work was performed at the Nanostructure Research Center, which is supported by the Fundamental Research Funds for the Central Universities (2021III016GX).

## Author contributions

W.J. and Z.Y.F. proposed the idea and designed the project. Z.Y.F. supervised the research. H.Y.X. and W.J. carried out most experiments,

analyzed data and wrote the manuscript. J.W.J. and J.L.L. helped the SEM-TKD characterization. J.S.W. and R.H.Y. carried out TEM experiments. W.M.W. and J.Y.Z. developed the theoretical model. B.T.T. and J.Z. performed crystal structure simulations and finite element analysis. H.W. and F.Z. helped in the revision. All authors discussed the results and revised the manuscript.

## Competing interests

The authors declare no competing interests.
