## [Peer Review File · Nature Communications]

Contribution of boundary non-stoichiometry to the lower-temperature plasticity in high-pressure sintered boron carbideREVIEWER COMMENTS

Reviewer #1 (Remarks to the Author):

Review on Manuscript Nature Communications 409914_0_merged_1674146451

“Title: Contribution of boundary non-stoichiometry to the lower-temperature plasticity in high-pressure sintered boron carbide

Authors: Zhengyi Fu * et all

This manuscript shows the view on the possibility to improve the brittle ceramic plasticity while (simultaneously) maintaining the high-temperature strength through the classical strategy. Authors propose to do this via decreasing grain size (of bulk boron carbide) to several nanometers and/or adding a ductile binder phase.

Authors claim that the plasticity of fully dense boron carbide has been greatly enhanced due to the boundary non-stoichiometry induced by high-pressure sintering technology. According to the authors' study, this (these) effect(s) decreased the plastic deformation temperature of B₄C by 200 °C compared to that of conventionally-sintered specimens. Promoted grain boundary diffusion is found to enhance grain boundary sliding, which dominates the lower-temperature plasticity. In addition, the as-produced specimen maintained extraordinary strength before the occurrence of plasticity. The study provides a new and efficient strategy by boundary chemical change to facilitate the plasticity of ceramic materials.

(1) The first sentence of the Abstract is very introductory. I am not sure it should be in the Abstract.

In Introduction

(2) Lines 35-36, page 2: How about the ref. <https://doi.org/10.1016/j.scriptamat.2021.114487>?

(3) Lines 41-42, page 2: What should be the particle size of boron carbide powder? What powder was used by the authors? This should be clearly specified in the manuscript.

Authors should clearly emphasize the novelty of their work in comparison with existing (see below) studies!

For this they have to refer to the following, recently published studies:

(4) “In Situ Observation of Fracture along Twin Boundaries in Boron Carbide”. Adv. Mater. 2022, 2204375 <https://doi.org/10.1002/adma.202204375>

(5) “Ultra-high temperature flexure and strain driven amorphization in polycrystalline boron carbide bulks”, Scripta Materialia 210 (2022) 114487 <https://doi.org/10.1016/j.scriptamat.2021.114487>

(6) “Plastic deformation in silicon nitride ceramics via bond switching at coherent interfaces” Science 378, 371-376 (2022) DOI: 10.1126/science.abq7490

In Results

(7) Lines 87-97, page 3. Authors claim a gradual increase of flexural strength for boron carbide from

room temperature to 1400 C from an average of 608 MPa to 820 MPa. The authors report some gradual decrease in strength to 600 MPa at about 1600 C. The further surprising drastic reduction in flexural strength to 385 MPa at 1700 C is totally unusual for B4C ceramic. Fig. 9 in “Room and high-temperature flexural failure of spark plasma sintered boron carbide”, *Ceramics International*, 42 7001-7013 (2016), <http://dx.doi.org/10.1016/j.ceramint.2016.01.088> Shows clear evidence of deformation resistivity of B4C at the temperature range of 25 – 1600 C. Further heating (flexural tests at higher temperatures leads to keeping of same high flexural strength or even to its increasing. Mechanism of ultra-high temperature flexure and strain-driven amorphization in polycrystalline boron carbide consolidated by spark plasma sintering method well analyzed in last year’s publication *Scripta Materialia* 210 (2022) 114487. The temperature dependence of the flexural strength of sintered boron carbide was studied. At temperatures above 2000oC, B4C showed an ultrahigh flexural strength far exceeding 1GPa (see Table 1) which was accompanied by a change in the deformation mechanism from brittle fracture to plastic deformation. Scanning transmission electron microscopy (STEM) observations revealed that the amorphization can be of microstructural origin for the observed plasticity in B4C at temperatures above 2000 °C and a strain rate of $3 \cdot 10^{-3} \text{ s}^{-1}$. The amorphization occurs inside the severely deformed grains. Flexural tests below 2000 °C provided evidence for the formation of stacking faults or dislocations, which are ordinary defects after the flexural tests. The results at 2000 °C suggest that the magnitude of the tensile stresses imposed on the B4C grains during deformation in flexure and the total strain transferred to a ceramic during the deformation process play the dominant role in the crystalline-amorphous transformation. Depending on the loading rate B4C ceramic showed 1000 - 8400 MPa (mean 1830 MPa) strength at a loading rate of 0.5 mm/min (the stress-strain curve was plastic). Increasing of loading rate to 2.5 mm/min leads to plastic deformation with a mean strength of 1250 MPa. Even very rapid deformation (in an elastic manner) at 25 mm/min has resulted in the same mean strength as was previously reported by these authors for room temperature (675 MPa) which clearly confirms the deformation resistivity of boron carbide under the temperature range of 25 – 2000 C. Authors must clearly refer to this work (“Ultra-high temperature flexure and strain driven amorphization in polycrystalline boron carbide bulks”, *Scripta Materialia* 210 (2022) 114487 <https://doi.org/10.1016/j.scriptamat.2021.114487>)

(8) Lines 95-97. This sentence is not UpToDate. Authors have to compare their results with the above-mentioned works.

In Methods

(9) Page 13 What was the 3% of impurities in Mudanjiang Diamond Boron Carbide Co., Ltd., China?

(10) What standard was used for the preparation (cutting) of test bars for three-point flexural testing?

The size of the test bars for the three-point test (2.2 mm × 3.8 mm × 25 mm) is a little bit strange.

(11) Table 1 includes a lot of information about preparation and testing conditions. This table should include one more column with the flexural strength for each testing temperature and crosshead speed.

(12) References should be updated.

Reviewer #2 (Remarks to the Author):

The manuscript reported the lower-temperature plasticity of B4C because of boundary non-stoichiometry. It is even more surprising that the strength was maintained at a high-temperature. The topic of ceramic plasticity is an important research field, and the results are certainly very interesting. It's generally believed the ceramics with nano-sized grain could show some ductility or plasticity. But it's very difficult to get the dense nanoceramics, even in the situation where a nano-sized powder is used. Another way of adding ductile binder could affect the high-temperature strength. Although intrinsic brittleness is the Achilles' heel of ceramics, the development was sluggish in recent years. This work shows an exciting result from the study of grain boundary of B4C prepared under high pressure. It is well-known that the grain boundaries in non-oxides are even more important. There has been quite a lot of work on the structure of grain boundaries in oxides over the past few years, but nearly little on the non-oxides. It may initiate a new opportunity for achieving plastic ceramics. The manuscript is well-written and the authors show very comprehensive and convincing results using advanced high temperature testing equipment and electron microscopy. Therefore, I recommend "publication" after a minor revision. The specific comments are as followed:

1. The results in Figure 1d-g and e-m clearly show different fracture modes for the high-pressure-low-temperature and the low-pressure-high-temperature sintered samples. The authors are able to quantify the ratio of different fracture modes at elevated bending temperatures. That will help to understand the brittle-ductile transition.
2. The most exciting result in the present work is about the boundary non-stoichiometry on the lower-temperature plasticity, which is quite different from the generally recognized nanoceramics or binder strategies. Some more experiments are suggested to be added using B4C with smaller grain size. It will be more attractive and convincing if the high-pressure densified finer B4C could show the similar trend with even lower-temperature plasticity.
3. The deformation calculation shows $n=2.1$ (grain boundary sliding), which is different from the Ref. 37 using coarse-grained B4C ($n=3$, power law creep). Though it's widely accepted that finer grain size favors grain boundary sliding over the power law creep, the reviewer suggests to add extra work using finer B4C. If n is close to 2, the results of the present work will be confirmed.
4. The reviewer agrees with the interpretation that the particular boundaries imaged have different structures, and there are differences in the EELS signals. But there is still a lack of direct evidence for the yield stress or hardness difference between the grain and boundary in HPLT B4C. It is suggested to try line/map scanning for hardness or elements.
5. It is an interesting comparison involving Ref. 11, which tests the micro-pillar in flash-sintered TiO₂. In Ref. 11, the nanotwins formed from pre-existed point defects during the compressive tests and help to cause deformation. But in this work, the micro-twins are present before and after, which shows it does not appear to contribute very much. It might play an important role in inhibiting dislocation motion, giving high-yield strength. The authors are pleased to add to the discussion.

6. The yield stress of ceramics is about a third of the hardness. It's better to add this statement after the sentence "The yield stress of B₄C ceramics at high temperatures was positively correlated with the hardness..." on Page 11.

7. Minor typo: In Fig.2, the letters m, n, and o in Fig.2k should be modified into the same font as others.

8. The manuscript shows many microstructure results from advanced electron microscopy. It's better to show the value of scale bars in the figures if that is allowed by the journal. That will help to convenient reading.

Dear Reviewers,

Thank you so much for the careful reading and scientific comment!

Following are our reply to the Reviewer #1:

Q1. *The first sentence of the Abstract is very introductory. I am not sure it should be in the Abstract.*

Response:

We fully agree with the comment! The first sentence of the original Abstract has been eliminated. In the revised manuscript, the first sentence has been modified as “The improvement of non-oxide ceramic plasticity while maintaining the high-temperature strength is a great challenge through the classical strategy, which generally includes decreasing grain size to several nanometers or adding ductile binder phase.”

Q2. *In Introduction:*

Lines 35-36, page 2: How about the ref.
<https://doi.org/10.1016/j.scriptamat.2021.114487?>

Response:

Thank you so much for the recommendation! The results reported in the literature is so attractive and very important for our work on the lower temperature plasticity. The recommended paper and the research group’s other novel studies on the promotion of the ultra-high temperature strength for boron carbide and boron carbide-based composites have been added in the references in the revised manuscript as followed. In addition, the additional review according to this reference has also been provided in the revised manuscript.

The added references:

[7] Demirskyi, D., Sepehri-Amin, H., Suzuki, T. S., Yoshimi, K., Vasylykiv, O. Ultra-high temperature flexure and strain driven amorphization in polycrystalline boron carbide bulks, *Scripta Mater.* **210**, 114487 (2022).

[8] Vasylykiv, O., Demirskyi, D., Borodianska, H., Kuncser, A., Badica, P. High-temperature strength of boron carbide with Pt grain-boundary framework in situ synthesized during spark plasma sintering, *Ceram. Inter.* **46**, 9136-9144 (2020).

[9] Kuncser, A., Vasylykiv, O., Borodianska, H. et al. High bending strength at 1800 °C exceeding 1 GPa in TiB₂-B₄C composite. *Sci. Rep.* **13**, 6915 (2023).

[10] Vasylykiv, O., Demirskyi, D., Badica, P., Nishimura, T., Tok, A. I. Y., Sakka, Y., Borodianska, H. Room and high temperature flexural failure of spark plasma sintered boron carbide. *Ceram. Inter.* **42**, 7001-7013 (2016).

Q3. *Lines 41-42, page 2: What should be the particle size of boron carbide powder? What powder was used by the authors? This should be clearly specified in the manuscript.*

Response:

Thanks for the suggestion! The B₄C powder (97% purity, Mudanjiang Diamond Boron Carbide Co., Ltd., China) with a median particle size of 2.5 μm was used as the starting material in the present work. The information has been clearly specified in the Methods part in the revised manuscript.

Authors should clearly emphasize the novelty of their work in comparison with existing (see below) studies! For this they have to refer to the following, recently published studies:

Q4. *“In Situ Observation of Fracture along Twin Boundaries in Boron Carbide”.* *Adv. Mater.* 2022, 2204375 <https://doi.org/10.1002/adma.202204375>

Response:

Thanks for the recommendation! The recommended paper shows new observation of fracture behavior in perfect and twinned B₄C at room temperature. It has been added into the references in the revised manuscript. The novelty of this study relates to different temperatures and different behaviors with our research.

This reference first revealed the exact role of twin boundaries (TBs) in the tensile behaviors of B₄C at room temperature. The results indicated that amorphous bands and cracking were preferentially initiated at the TBs in B₄C under both indentation and tension loading. The cracks then propagated along the TBs, thus resulting in the fracture of B₄C. While our research mainly explored the plasticity deformation mechanism of high-pressure sintered B₄C related to grain boundary, at temperatures ranging from 1100°C to 2000 °C.

The added references:

[42] Li, P. H., Bu, Y. Q., Wang, L. Y., Wang, C., Huang, J. Q., Tong, K., Chen, Y. J., He, J. L., Zhao, Z. S., Xu, B., Liu, Z. Y., Gao, G. Y., Nie, A. M., Wang, H. T., Tian, Y. J. In situ observation of fracture along twin boundaries in boron carbide. *Adv. Mater.* 2204375 (2022).

Q5. “Ultra-high temperature flexure and strain driven amorphization in polycrystalline boron carbide bulks”, *Scripta Materialia* 210 (2022) 114487
<https://doi.org/10.1016/j.scriptamat.2021.114487>

Response:

Thanks for the recommendation! The paper has been added in the references and extra discussion has been added in the Introduction part of the revised manuscript. The paper is very important, which concentrated on the different novelties from our research.

This paper shows impressive results mainly focusing on the high flexure strength at ultra-high temperature above 2000 °C, and the strain driven amorphization of fully dense SPS-ed B₄C ceramics with additions of 25 wt.% B and 0.5 mol.% Si. Though the main purpose of boron addition is to balance the boron to carbon ratio and bond the carbon coming from the SPS die, it may also change the final composition and then affect the plastic performance. In addition, Si segregation could be clearly observed at the B₄C grain boundary before and after the deformed ceramics.

The paper revealed that the amorphization occurs inside of the severely deformed grains at 2000 °C. The magnitude of the tensile stresses imposed on the B₄C grains during deformation in flexure and the total strain transferred to a ceramic during the deformation process and played the dominant role in the crystalline-amorphous transformation.

Our research mainly concentrated on how to induce the ceramic plasticity at relative lower temperature. The microstructure evolution and plastic deformation mechanism studies mainly focused on the HPLT-B₄C sample tested at 1700 °C and the LPHT-B₄C tested at 1900 °C.

The added references:

[7] Demirskyi, D., Sepehri-Amin, H., Suzuki, T. S., Yoshimi, K., Vasylykiv, O. Ultra-high temperature flexure and strain driven amorphization in polycrystalline boron carbide bulks, *Scripta Mater.* **210**, 114487 (2022).

Q6. “Plastic deformation in silicon nitride ceramics via bond switching at coherent interfaces” *Science* 378, 371-376 (2022) DOI: 10.1126/science.abq7490

Response:

This recommended article reported very important research on the plastic deformation of ceramics. We have added it in the references. It introduces a different approach and different sample dimension from our research to realize low temperature plasticity.

The paper presents a new approach defined as “bonding switching” for the deformable covalently bonded Si₃N₄. The method is very strict that require the “bond breaking in covalent bonds occurs in a very small volume together with immediate healing by new bond formations”. They found a specific dual-phase α/β-Si₃N₄ ceramic with specific coherent interface, and realized the requirement through the stress-induced β→α phase transformation at this interface. In addition, it’s kind of local plastic, because the dimensions of micro-compressive tested nanopillar is about Φ 350 nm × 350 nm, which only contains very limited grains.

Our research shows the macro-mechanical properties of a bulk B₄C sample. The plasticity is associated with all the grain boundaries under stress. The reference authors also admitted that better fabrication method should be required to obtain bulk-scale deformable Si₃N₄ ceramics with dual-phase grand structure and coherent interfaces.

The added references:

[11] Zhang, J., Liu, G. H., Cui, W., Ge, Y. Y., Du, S. M., Gao, Y. X., Zhang, Y. Y., Li, F., Chen, Z. L., Du, S. X., Chen, K. X. Plastic deformation in silicon nitride ceramics via bond switching at coherent interfaces. *Science* **378**, 371-376 (2022).

In Results

Q7. Lines 87-97, page 3. Authors claim a gradual increase of flexural strength for boron carbide from room temperature to 1400 °C from an average of 608 MPa to 820 MPa. The authors report some gradual decrease in strength to 600 MPa at about 1600 °C. The further surprising drastic reduction in flexural

strength to 385 MPa at 1700 C is totally unusual for B₄C ceramic. Fig. 9 in “Room and high-temperature flexural failure of spark plasma sintered boron carbide”, *Ceramics International*, 42 7001-7013 (2016), <http://dx.doi.org/10.1016/j.ceramint.2016.01.088> Shows clear evidence of deformation resistivity of B₄C at the temperature range of 25-1600 °C. Further heating (flexural tests at higher temperatures leads to keeping of same high flexural strength or even to its increasing. Mechanism of ultra-high temperature flexure and strain-driven amorphization in polycrystalline boron carbide consolidated by spark plasma sintering method well analyzed in last year’s publication *Scripta Materialia* 210 (2022) 114487. The temperature dependence of the flexural strength of sintered boron carbide was studied. At temperatures above 2000 °C, B₄C showed an ultrahigh flexural strength far exceeding 1GPa (see Table 1) which was accompanied by a change in the deformation mechanism from brittle fracture to plastic deformation. Scanning transmission electron microscopy (STEM) observations revealed that the amorphization can be of microstructural origin for the observed plasticity in B₄C at temperatures above 2000 °C and a strain rate of $3 \cdot 10^{-3} \text{s}^{-1}$. The amorphization occurs inside the severely deformed grains. Flexural tests below 2000 °C provided evidence for the formation of stacking faults or dislocations, which are ordinary defects after the flexural tests. The results at 2000 °C suggest that the magnitude of the tensile stresses imposed on the B₄C grains during deformation in flexure and the total strain transferred to a ceramic during the deformation process play the dominant role in the crystalline-amorphous transformation. Depending on the loading rate B₄C ceramic showed 1000 - 8400 MPa (mean 1830 MPa) strength at a loading rate of 0.5 mm/min (the stress-strain curve was plastic). Increasing of loading rate to 2.5 mm/min leads to plastic deformation with a mean strength of 1250 MPa. Even very rapid deformation (in an elastic manner) at 25 mm/min has resulted in the same mean strength as was previously reported by these authors for room temperature (675 MPa) which clearly confirms the deformation resistivity of boron carbide under the temperature range of 25-2000 °C. Authors must clearly refer to this work (“Ultra-high temperature flexure and strain driven amorphization in polycrystalline boron carbide bulks”, *Scripta Materialia* 210 (2022) 114487 <https://doi.org/10.1016/j.scriptamat.2021.114487>)

Response:

Thank you so much for the professional comment! The recommended researches are very important and have been added into the references in our revised manuscript.

In the article "Room and high-temperature flexural failure of spark plasma sintered boron carbide, *Ceramics International*, 42 7001-7013 (2016)", the raw grain size is 4 μm , and grain growth should occur at high sintering temperature. The process is similar to our study's control sample that sintered at high temperature with low pressure (LPHT-B₄C). The bending strength shows a gradual increase from room temperature to 1600 °C, and even maintain a very high strength (767 MPa), which agree well with the A40 sample in Fig. 9 in the recommend paper. The tiny difference may also associate with the fine grain-boundary metal Pt framework in the recommended paper. The fully dense HPLT-B₄C without grain growth in our study was obtained under high pressure at low temperature. The plasticity of B₄C was enhanced due to the boundary non-stoichiometry induced by high-pressure sintering technology and shows lower temperature plasticity at 1600 °C.

For the article "Scripta Materialia 210 (2022) 114487", the reason for the differences in deformation mechanism and flexure performance are as followed.

1. The grain size is different. The grain size of raw B₄C powder from "Scripta Materialia 210 (2022) 114487" is twice as large as that in our study. The grain should grow during high temperature sintering. So, the larger grain size could improve the plasticity temperature of B₄C ceramics.

2. The plasticity temperature is different. In the reference, B₄C ceramics occurred plasticity at 2000 °C, the amorphization can be of microstructural origin for the observed plasticity in B₄C at temperatures above 2000 °C. In our study, the plastic temperature is reduced and amorphization was not observed in the lower temperature testing.

3. The novelty related to the plasticity mechanism is different. In the reference, the dislocation motion was the dominating plastic deformation. In our study, grain boundary sliding is proved as the dominate mechanism with much smaller grain size and boundary non-stoichiometry. It may cause the differences of mechanical performance.

4. Another possible reason is the composition. The reference used raw

material with boron addition of 25 wt.%, and 0.5 mol.% of Si was presented in the as received B₄C powder. Though the main purpose of boron addition is to balance the boron to carbon ratio and bond the carbon coming from the SPS die, it may also change the final composition and then affect the plastic performance. In addition, Si segregation could be clearly observed at the B₄C grain boundary before and after the deformed ceramics. A latest and novel literature “High bending strength at 1800 °C exceeding 1 GPa in TiB₂-B₄C composite. *Sci. Rep.* **13**, 6915 (2023)” shows a similar relatively linear strengthening region beyond 10% strain after plasticity appeared. The paper about TiB₂-B₄C composite has also been added into the references.

To sum up, the paper “*Scripta Materialia* 210 (2022) 114487” is very attractive in the ultra-high temperature flexure performance study of B₄C ceramics. While our study concentrates on the different novelties and provides a new and efficient strategy by boundary chemical change to facilitate the plasticity of ceramics.

The added references:

[7] Demirskyi, D., Sepehri-Amin, H., Suzuki, T. S., Yoshimi, K., Vasylykiv, O. Ultra-high temperature flexure and strain driven amorphization in polycrystalline boron carbide bulks, *Scripta Mater.* **210**, 114487 (2022).

[9] Kuncser, A., Vasylykiv, O., Borodianska, H. et al. High bending strength at 1800 °C exceeding 1 GPa in TiB₂-B₄C composite. *Sci. Rep.* **13**, 6915 (2023).

[10] Vasylykiv, O., Demirskyi, D., Badica, P., Nishimura, T., Tok, A. I. Y., Sakka, Y., Borodianska, H. Room and high temperature flexural failure of spark plasma sintered boron carbide. *Ceram. Inter.* **42**, 7001-7013 (2016).

Q8. Lines 95-97. This sentence is not UpToDate. Authors have to compare their results with the above-mentioned works.

Response:

Thank you for the carefully reading! The sentence in Lines 95-97 has been updated into “However, after the brittle-ductile transition temperature was reached, the plastic yield stress decreased with elevated temperatures [7-10, 40, 42].” The above-mentioned works and some other paper have been referred in the revised manuscript.

[7] Demirskyi, D., Sepehri-Amin, H., Suzuki, T. S., Yoshimi, K., Vasylykiv, O.

Ultra-high temperature flexure and strain driven amorphization in polycrystalline boron carbide bulks, *Scripta Mater.* **210**, 114487 (2022).

[8] Vasylykiv, O., Demirskyi, D., Borodianska, H., Kuncser, A., Badica, P. High-temperature strength of boron carbide with Pt grain-boundary framework in situ synthesized during spark plasma sintering, *Ceram. Inter.* **46**, 9136-9144 (2020).

[9] Kuncser, A., Vasylykiv, O., Borodianska, H. et al. High bending strength at 1800 °C exceeding 1 GPa in TiB₂-B₄C composite. *Sci. Rep.* **13**, 6915 (2023).

[10] Vasylykiv, O., Demirskyi, D., Badica, P., Nishimura, T., Tok, A. I. Y., Sakka, Y., Borodianska, H. Room and high temperature flexural failure of spark plasma sintered boron carbide. *Ceram. Inter.* **42**, 7001-7013 (2016).

[29] Moshtaghioun, B.M., García, D.G., Rodríguez, A.D., Padture, N.P. High-temperature creep deformation of coarse-grained boron carbide ceramics. *J. Eur. Ceram. Soc.* **35**, 1423-1429 (2015).

[30] Abzianidze, T.G., Eristavi, A.M., Shalamberidze, S.O. Strength and creep in boron carbide (B₄C) and aluminum dodecaboride (α -AlB₁₂). *J. Solid State Chem.* **154** 191-193 (2000).

In Methods

Q9. Page 13 What was the 3% of impurities in Mudanjiang Diamond Boron Carbide Co., Ltd., China?

Response:

According to the supplier's information, the 3% impurities include 0.7% N, 1.8% O, 0.1% Fe, 0.15% Si, 0.05% Al and 0.2% other elements (Ca, Cr, Mg, Mn, Ni, Ti, W). It should be pointed out that for boron carbide, it's very hard to get high-purity (>99%) and fine powder because of the synthetic process.

Q10. What standard was used for the preparation (cutting) of test bars for three-point flexural testing? The size of the test bars for the three-point test (2.2 mm × 3.8 mm × 25 mm) is a little bit strange.

Response:

Thanks for the carefully reading! The preparation of sample bars for three-point flexural testing was determined by the Chinese Standard GBT 14390-2008: "Fine ceramics (advanced ceramics, advanced technical ceramics) - Test

method for flexural strength of monolithic ceramics at elevated temperature”. According to the standard and available literatures, the dimension is not so strictly restrained. It could be changed to keep a small deflection in bending, according to the real situation about the span and the material. Because of a short span of 16 mm in the present work, the test bar with the identical dimension of 2.2 mm×3.8 mm×25mm was used.

Q11. *Table 1 includes a lot of information about preparation and testing conditions. This table should include one more column with the flexural strength for each testing temperature and crosshead speed.*

Response:

We appreciate your kind advice! The flexural strength for each testing temperature and crosshead speed has been added in the new column in Table 1 as followed.

Table 1. Summary of Sintering, Test Conditions and Flexural Strength of B₄C Ceramics

Samples	Sintering Temperature (°C)	Sintering Pressure (MPa)	Test Temperature (°C)	Crosshead Speed (mm/min)	Flexure Strength (MPa)
HPLT-B ₄ C-1100-0.5	1800	80	1100	0.5	617
HPLT-B ₄ C-1300-0.5	1800	80	1300	0.5	772
HPLT-B ₄ C-1400-0.5	1800	80	1400	0.5	823
HPLT-B ₄ C-1500-0.5	1800	80	1500	0.5	801
HPLT-B ₄ C-1600-0.5	1800	80	1600	0.5	612
HPLT-B ₄ C-1700-1	1800	80	1700	1	454
HPLT-B ₄ C-1700-0.5	1800	80	1700	0.5	320
HPLT-B ₄ C-1700-0.25	1800	80	1700	0.25	232
HPLT-B ₄ C-1800-0.5	1800	80	1800	0.5	130
HPLT-B ₄ C-1900-0.5	1800	80	1900	0.5	39
HPLT-B ₄ C-2000-0.5	1800	80	2000	0.5	17
LPHT-B ₄ C-1300-0.5	2100	20	1300	0.5	588
LPHT-B ₄ C-1500-0.5	2100	20	1500	0.5	772
LPHT-B ₄ C-1600-0.5	2100	20	1600	0.5	821
LPHT-B ₄ C-1700-0.5	2100	20	1700	0.5	767
LPHT-B ₄ C-1800-0.5	2100	20	1800	0.5	703
LPHT-B ₄ C-1900-0.5	2100	20	1800	0.5	143
LPHT-B ₄ C-1950-1	2100	20	1950	1	101
LPHT-B ₄ C-1950-0.5	2100	20	1950	0.5	71
LPHT-B ₄ C-1950-0.25	2100	20	1950	0.25	55
LPHT-B ₄ C-2000-0.5	2100	20	2000	0.5	36

Q12. *References should be updated.*

Response:

We are so sorry for the negligence of the latest references about the high temperature strength of boron carbide and some other non-oxide ceramics. According to the kind suggestion, the updated references have been added in the revised manuscript as followed. In addition, the additional literature review has added in the paper as well.

[7] Demirskyi, D., Sepehri-Amin, H., Suzuki, T. S., Yoshimi, K., Vasylykiv, O. Ultra-high temperature flexure and strain driven amorphization in polycrystalline boron carbide bulks, *Scripta Mater.* **210**, 114487 (2022).

[8] Vasylykiv, O., Demirskyi, D., Borodianska, H., Kuncser, A., Badica, P. High-temperature strength of boron carbide with Pt grain-boundary framework in situ synthesized during spark plasma sintering, *Ceram. Inter.* **46**, 9136-9144 (2020).

[9] Kuncser, A., Vasylykiv, O., Borodianska, H. et al. High bending strength at 1800 °C exceeding 1 GPa in TiB₂-B₄C composite. *Sci. Rep.* **13**, 6915 (2023).

[10] Vasylykiv, O., Demirskyi, D., Badica, P., Nishimura, T., Tok, A. I. Y., Sakka, Y., Borodianska, H. Room and high temperature flexural failure of spark plasma sintered boron carbide. *Ceram. Inter.* **42**, 7001-7013 (2016).

[11] Zhang, J., Liu, G. H., Cui, W., Ge, Y. Y., Du, S. M., Gao, Y. X., Zhang, Y. Y., Li, F., Chen, Z. L., Du, S. X., Chen, K. X. Plastic deformation in silicon nitride ceramics via bond switching at coherent interfaces. *Science* **378**, 371-376 (2022).

[42] Li, P. H., Bu, Y. Q., Wang, L. Y., Wang, C., Huang, J. Q., Tong, K., Chen, Y. J., He, J. L., Zhao, Z. S., Xu, B., Liu, Z. Y., Gao, G. Y., Nie, A. M., Wang, H. T., Tian, Y. J. In situ observation of fracture along twin boundaries in boron carbide. *Adv. Mater.* 2204375 (2022).

Following are our reply to the Reviewer 2:

Q1. *The results in Figure 1d-g and e-m clearly show different fracture modes for the high-pressure-low-temperature and the low-pressure-high-temperature sintered samples. The authors are able to quantify the ratio of different fracture modes at elevated bending temperatures. That will help to understand the brittle-ductile transition.*

Response:

Thank you so much for the suggestion! We have quantified the ratio of different fracture mode at elevated bending temperatures from the fracture surface images, for both the high-pressure-low-temperature sintered B₄C (HPLT-B₄C) and low-pressure-high-temperature B₄C (LPHT-B₄C). It should be pointed out that the statistical result is relatively rough. The results are shown in the bar graph in **Fig. R2** as followed.

According to the image analysis, the HPLT-B₄C ceramics possess 95% transgranular and 5% intergranular fractures at 1100 °C, 89% transgranular and 11% intergranular fractures at 1400 °C, and 15% transgranular and 85% intergranular fractures at 1600 °C. The LPHT-B₄C possess 97% transgranular and 3% intergranular fractures at 1300 °C, 93% transgranular and 7% intergranular fractures at 1600 °C, and 91% transgranular and 9% intergranular fractures at 1800 °C.

Fig.R2 Quantified ratio of different fracture modes at elevated bending temperatures for the as-sintered B₄C ceramics. **(a)** HPLT-B₄C. **(b)** LPHT-B₄C

Q2. *The most exciting result in the present work is about the boundary non-stoichiometry on the lower-temperature plasticity, which is quite different from the generally recognized nanoceramics or binder strategies. Some more experiments are suggested to be added using B₄C with smaller grain size. It will be more attractive and convincing if the high-pressure densified finer B₄C could show the similar trend with even lower-temperature plasticity.*

Response:

We fully agree with the reviewer's viewpoint that using smaller raw powder may get more attractive and convincing results if the high-pressure densified

finer B₄C could show the similar trend with even lower-temperature plasticity.

According to the reviewer's suggestion. The finer raw B₄C powder with grain size of 1.2 μm from the same supplier was used in the extra experiments. The 1700 °C / 80 MPa HPLT-ed fine B₄C ceramics with grain size of 1.2 μm and 2000 °C / 20 MPa LPHT-ed fine B₄C ceramics with grain size of 2.4 μm were obtained after the optimized sintering parameters.

The high-temperature flexural test in **Fig. R3** clearly reveals that the HPLT-ed fine B₄C could show plasticity at lower temperature of 1500 °C, while LPHT-ed fine B₄C show plasticity at higher temperature of 1700 °C. The results showed the same trend as the manuscript, and confirmed the positive effect of the boundary non-stoichiometry on the lower-temperature plasticity. In addition, though the LPHT-ed fine B₄C ceramics possess finer grain size (2.4 μm) than the HPLT-ed original B₄C ceramics (2.5 μm), the plasticity temperature is still much higher. The results also reveal that boundary non-stoichiometry by high pressure greatly favor to lower-temperature plasticity.

Fig. R3 High-temperature flexural test at a constant strain rate for B₄C ceramics sintered from finer raw powder (1.2 μm). **(a)** HPLT-ed fine B₄C ceramics with grain size of 1.2 μm. **(b)** LPHT-ed fine B₄C ceramics with grain size of 2.4 μm.

Q3. The deformation calculation shows $n=2.1$ (grain boundary sliding), which is different from the Ref. 37 using coarse-grained B₄C ($n=3$, power law creep).

Though it's widely accepted that finer grain size favors grain boundary sliding over the power law creep, the reviewer suggests to add extra work using finer B₄C. If *n* is close to 2, the results of the present work will be confirmed.

Response:

We fully agree with the reviewer's comment. Extra work using finer B₄C powder with particle size of 1.2 μm has been added. As shown in the reply to Question 2#, the grain size of HPLT-ed fine B₄C was 1.2 μm. The sample was tested at 1700 °C under various strain rates. The crosshead speeds are 0.25 mm/min, 0.5 mm/min and 1 mm/min, respectively. The strain-stress curves and deformation calculation of *n* value are shown in **Fig. R4**. The results indicate *n*=2.04 for the HPLT-ed fine B₄C, which has confirmed that the deformation mechanism is grain boundary sliding in the HPLT-B₄C in the original manuscript.

It should be pointed out that because of the extra experiment used finer B₄C powder, the value of *n* is closer to 2 than the original manuscript. It makes sense and agrees with the widely accepted viewpoint "finer grain size favors grain boundary sliding over the power law creep" as the reviewer agreed in the comment.

Fig. R4 High-temperature plastic deformation behavior and mechanisms of HPLT-ed fine B₄C with grain size of 1.2 μm. **(a)** Stress-strain curves, **(b)** strain rate dependence of yield stress.

Q4. The reviewer agrees with the interpretation that the particular boundaries imaged have different structures, and there are differences in the EELS signals. But there is still a lack of direct evidence for the yield stress or hardness difference between the grain and boundary in HPLT B₄C. It is suggested to try line/map scanning for hardness or elements.

Response:

According to the reviewer's suggestion, the extra experiment of microhardness mapping has been added. An in-situ Nanoindentation Testing System (FT-NMT04, FemtoTools AG, Switzerland) was employed. The nanoindentation mapping was done in displacement-controlled mode with a constant depth of 25 nm. An ultra-fine indent spacing of 250 nm was adopted to cover an area of $4\ \mu\text{m} \times 3\ \mu\text{m}$. The high-resolution nanoindentation mapping results were shown in **Fig. R5**.

It clearly reveals that, in the HPLT-B₄C ceramics, the hardness near grain boundary is relative lower than that inside the grain. The results have supported the direct evidence for the hardness difference between the grain and boundary in HPLT-B₄C.

Fig. R5 Microhardness mapping of HPLT-B₄C ceramics. (a) The SEM image of polished surface. (b-c) The selected area and the corresponding hardness mapping image.

Q5. *It is an interesting comparison involving Ref. 11, which tests the micro-pillar in flash-sintered TiO₂. In Ref. 11, the nanotwins formed from pre-existed point defects during the compressive tests and help to cause deformation. But in this work, the micro-twins are present before and after, which shows it does not appear to contribute very much. It might play an important role in inhibiting dislocation motion, giving high-yield strength. The authors are pleased to add to the discussion.*

Response:

Thanks for the suggestion!

In Ref. 11, nanoscale twin boundary is considered to play an important role in the deformation at 400 °C for flash sintered TiO₂. In our study, both the

mathematical model and microstructure evolution indicate the grain boundary sliding was the dominating mechanism. While twinning serves in coordination with plastic deformation for B₄C at 1700 °C -2000 °C. From the GPA analysis in Fig.3, the twin boundary among B₄C ceramics exerts the compressive prestress, which could enhance the mechanical properties. In addition, the twin boundary may also play a role of plastic accommodation for B₄C ceramics. According to the reference “Xiao et al., Dislocation behaviors in nanotwinned diamond. *Sci. Adv.* **4**: eaat8195 (2018)”, the reaction of dislocation and twin plane could improve plasticity and strength. The extra reference has been added into the discussion.

Q6. *The yield stress of ceramics is about a third of the hardness. It's better to add this statement after the sentence “The yield stress of B₄C ceramics at high temperatures was positively correlated with the hardness...” on Page 11.*

Response:

Thank you so much for the professional comment! According to the suggestion, the sentence “The yield stress of ceramics is estimated to be a third of the hardness.” has been added on Page 11 in the revised manuscript.

Q7. Minor typo: In Fig.2, the letters m, n, and o in Fig.2k should be modified into the same font as others.

Response:

Thank you so much for the carefully reading! The letters m, n, and o in Fig. 2k have been modified into the same font (Arial) as others. In addition, all the figures have been carefully checked to keep all the letter in the uniform font.

Q8. *The manuscript shows many microstructure results from advanced electron microscopy. It's better to show the value of scale bars in the figures if that is allowed by the journal. That will help to convenient reading.*

Response:

We fully agree with the comment! According to your suggestion, the value and unit of the scale bars in all the figures have been added in the revised manuscript and supplementary information. Of course, we will definitely follow the journal's publishing rule.

REVIEWERS' COMMENTS

Reviewer #1 (Remarks to the Author):

Revising the manuscript authors made their work clear and sharp.

What are the noteworthy results?

It was shown that the plasticity of boron carbide has been enhanced due to the boundary non-stoichiometry induced by high-pressure sintering. The effect decreased the plastic deformation temperature by 200 °C compared to that of conventionally-sintered specimens. Promoted grain boundary diffusion enhances grain boundary sliding, which dominates the lower-temperature plasticity. Boron carbide maintained high strength before the occurrence of plasticity. The study provides an efficient strategy by boundary chemical change to facilitate the low-temperature plasticity of ceramic materials.

Will the work be of significance to the field and related fields?

Definitely will.

How does it compare to established literature? If the work is not original, please provide relevant references.

The revised manuscript clearly shows the novelty and originality of this work.

Does the work support the conclusions and claims, or is additional evidence needed?

No additional evidence is needed. The paper may now be published as is.

Are there any flaws in the data analysis, interpretation, and conclusions? Do these prohibit publication or require revision?

No further revision is necessary.

Is the methodology sound? Does the work meet the expected standards in your field?

The work is very important, sounds, and meets the highest standards in our field.

Is there enough detail provided in the methods for the work to be reproduced?

Yes, the description is very detailed and the reproducibility of the results has no doubt.

Reviewer #2 (Remarks to the Author):

The authors have addressed the comments well for the revise manuscript, and the manuscript is recommended for an "Accept".